# Decoupling both local and global abundance from global range size, challenging the abundance-occupancy relationship in birds

Shinichi Nakagawa[1,2]*, William K Cornwell[1†], Corey T Callaghan[3]*†

[1]Evolution & Ecology Research Centre, Centre for Ecosystem Science, and School of Biological, Earth and Environmental Sciences, University of New South Wales, Sydney, Australia; [2]Theoretical Sciences Visiting Program, Okinawa Institute of Science and Technology Graduate University, Onna, Japan; [3]Department of Wildlife Ecology and Conservation, Fort Lauderdale Research and Education Center, University of Florida, Davie, United States

*For correspondence:
snakagaw@ualberta.ca (SN);
c.callaghan@ufl.edu (CTC)

†These authors contributed equally to this work

## eLife assessment

This study offers a **useful** discussion of the well-accepted abundance-occupancy relationship in macroecology. While using the ebird large dataset to revisit the theme is interesting, multiple unresolved confounding factors exist, leaving the results **inadequate** to overturn the repeatedly confirmed abundancy-occupancy relationship.

**Abstract** In macroecology, a classic empirical observation has been positive relationships between local abundance and species' range, known as the abundance-occupancy relationships (AORs). The existence of this empirical relationship has informed both theory development and applied questions. Notably, the spatial neutral model of biodiversity predicts AORs. Yet, based on the largest known meta-analysis of 16,562,995 correlations from ~3 billion bird observations, this relationship was indistinguishable from zero. Further, in a phylogenetic comparative analysis, species range had no predictive power over the global mean abundance of 7464 bird species. We suggest that publication and confirmation biases may have created AORs, an illusion of a 'universal' pattern. This nullification highlights the need for ecologists to instigate a credibility revolution like psychology, where many classic phenomena have been nullified.

## Introduction

A positive interspecific relationship between abundance and distribution –abundance-occupancy relationships (AORs) – is considered one of the most general and robust patterns in ecology (**Blackburn et al., 2006**; **Gaston, 1996**; **Gaston et al., 1997b**; **Ten Caten et al., 2022**). Sometimes referred to as a macroecological law (**Gaston and Blackburn, 1999b**; **Lawton, 1999**), the AOR asserts that empirically locally abundant species tend to be widely distributed, and conversely, locally rare species tend to be geographically restricted in their range. The mechanism driving this relationship was never proven, and it remains unresolved why species distribution should affect per-unit-area abundance (or vice versa). Nonetheless, the existence of a pervasive AOR has underpinned many practical applications in ecology and conservation (**Gaston, 1999a**), e.g., setting harvest rates for fisheries (**Swain**

*and Morin, 1996*), managing invasive species by restricting expansion rather than local elimination, and identifying species at high risk of extinction in biodiversity inventories such as the IUCN Red List Criteria (*Gaston et al., 2000*). Given the increasing human-induced land-use changes in the Anthropocene (*Lewis and Maslin, 2015*), concomitantly with increasing debate about global biodiversity change (*Leung et al., 2020*), fully understanding the relationship between abundance and range size is increasingly important.

Many plausible biological mechanisms have been proposed for AORs, yet none of them has unequivocal support (*Gaston et al., 1997b*; *Ten Caten et al., 2022*; *Wilson, 2008*; *Borregaard and Rahbek, 2010*). Among all mechanisms, it is noteworthy that a spatially explicit neutral model of biodiversity and biogeography can generate AORs (*Bell, 2001*; *Hubbell, 1997*). Specifically, this macro-ecological 'null' model can produce a positive correlation between species range (or occupancy) and their per-unit-area local, as well as total global abundance. This observation, in turn, supports the utility of neutral theory as a null model of community and macroecology (*Bell, 2001*). Although neutral theory may provide a biological null model, an additional null hypothesis is that AOR does not exist. Indeed, sampling bias can create AORs because locally rare species are more likely to be missed, resulting in an underestimation of range size or occupancy, thereby generating a positive relationship (*Borregaard and Rahbek, 2010*; *Bock and Ricklefs, 1983*). Yet, this sampling explanation has long been discarded as a plausible mechanism leading to observed patterns (*Gaston, 1996*; *Gaston et al., 1997b*; *Blackburn and Gaston, 2009*). This is because of substantial empirical evidence for positive interspecific relationships, including a meta-analysis of 279 effect sizes with an overall effect of $r=0.58$ (or its Fisher's transformation: $Zr = 0.66$) in 2006 (*Blackburn et al., 2006*). It does not seem that sampling bias alone could explain this remarkably strong relationship.

Nonetheless, a large amount of variation does exist in empirical patterns of AORs, including strikingly negative relationships (*Wilson, 2008*; *Päivinen et al., 2005*; *Komonen et al., 2009*; *Kotiaho et al., 2009*). Some of the observed heterogeneity is likely to be due to different aspects of sampling, such as the number of species and spatial and temporal coverage (*Gaston et al., 1997b*; *Ten Caten et al., 2022*; *Wilson, 2008*). Also, other types of bias could generate artefactual AORs: namely 'confirmation bias', where sampling is prejudiced to support one's hypothesis, and 'publication bias', where statistically significant relationships are preferentially reported and published. Although both biases are widespread, including in ecological studies (*Holman et al., 2015*; *van Wilgenburg and Elgar, 2013*; *Yang et al., 2023*), no studies so far systematically considered or quantified both biases in the context of AORs (*Blackburn et al., 2006*). Furthermore, there has, until recently, been a lack of large and methodologically consistent data resources, therefore leaving a traditional meta-analytic approach as the best available option for testing the validity and generality of the AOR.

## Results

### A citizen science dataset to test AORs

Here, we use data from eBird – a global citizen science dataset aimed at counting birds – to quantify the relationship between local-scale (and global-scale) abundance and global-scale range size as a proxy for occupancy (AOR). This approach is similar to previous works (e.g. *Bock and Ricklefs, 1983*); they pointed out that the use of arbitrary cut-offs in many AOR studies can lead to artefactual positive AOR relationships. By examining this relationship across a global dataset, we aim to test whether the classic AOR pattern holds at a broader scale using citizen science data, which provides a more comprehensive spatial coverage than traditional studies. Previous AOR studies often focused on local or regional scales, defining occupancy within specific patches or habitats. In contrast, our approach uses global range size to explore how generalisable AOR patterns are when scaled up to global datasets, providing insights into whether the same positive relationship persists across diverse environments and species distributions.

Large citizen science datasets collected for non-hypothesis-driven purposes are not random samples (see *Callaghan et al., 2017*), but they have the advantage of avoiding biases such as confirmation and publication bias. Also, using the eBird dataset allows us to estimate heterogeneity due to sampling intensity (e.g. the duration of a sampling event directly influences the number of species recorded). Specifically, we can quantify how AOR will change in relation to increases in species richness

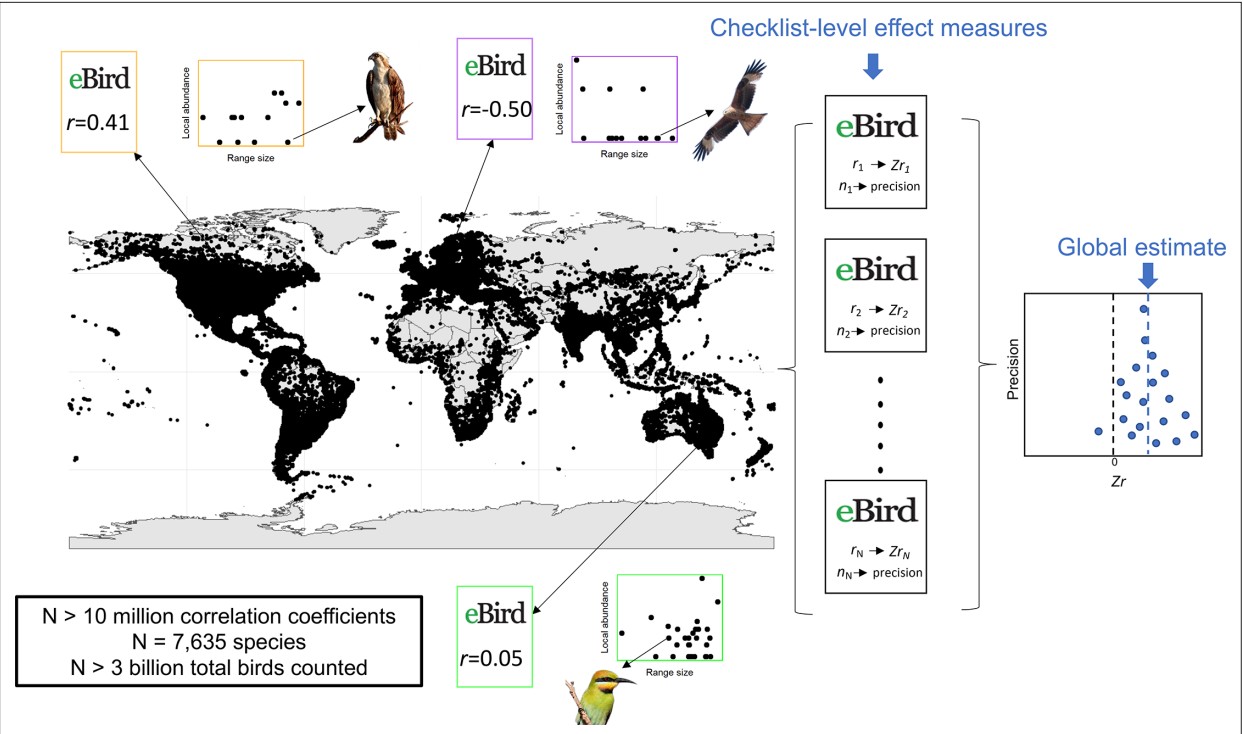

**Figure 1.** A conceptual overview of our methods. We aggregated individual eBird checklists across the world (shown on the map), represented by the three coloured insets which show the relationship between global range size (x-axis) and local abundance (y-axis) and the associated correlation value. We then aggregated these checklist level measures for 16,562,995 eBird checklists into the largest-ever meta-analysis to find the global-level relationship between global range size and local abundance.

and sampling duration, both of which are predicted to reduce the magnitude of AORs (*Gaston et al., 1997b*; *Ten Caten et al., 2022*; *Komonen et al., 2009*).

For occupancy, we use global range size not only because global range size should be relatively stable – 'local' range sizes for one species could vary dramatically – but also because different types of occupancy measures were deemed to contribute less to the observation heterogeneity (*Wilson, 2011*; *Steenweg et al., 2018*). Fortunately, for birds, a large database of global range sizes has already been compiled (*BirdLife International, 2023*). For abundance, we use two different measurements: local species counts and local mean density, as follows. First, we carry out the largest known meta-analysis by synthesising correlations between global range sizes of 7635 species and local species counts collected across 16,562,995 eBird checklists (resulting in 16,562,995 *Zr* values and corresponding sampling variances; *Figure 1*). These checklists all included counts of each species present and the duration of observation (hereafter, effort time).

Second, we conduct a phylogenetically controlled comparative analysis, regressing species range sizes on 7464 estimates of globally derived species' mean density, equivalent to mean local density (per 5-degree grid cell), estimated in earlier work (*Callaghan et al., 2021*) (see Materials and methods for more details). Given the different potential biases mentioned before, we expected a more modest relationship in relation to that of the previous meta-analysis (*r*=0.58; note that this relationship included many different taxa; if restricted to bird species, it was even stronger *r*=0.74 or *Zr* = 0.95) (*Blackburn and Gaston, 2009*). Also, although no such empirical evidence appears to exist, it seems feasible that in filling in an eBird checklist, some people may undercount common and widespread species while they may overcount rare and geographically restricted species. If this is the case, the relationship (AORs) could be further weakened. Yet, if such overcounting and undercounting were present, we expect it would introduce large heterogeneity into our dataset because that type of behaviour would not be consistent across all contributors, and they would sometimes result in negative AORs, increasing variability among the 16,562,995 *Zr* values.

## Overwhelming support against AOR

Surprisingly, the overall (aggregated) relationship between local abundance and global occupancy was near-zero ($r$=0.015), although this relationship was statistically significant due to our extremely large sample size (p=0.0005, $z$=2.805, $Zr$ = $b_{\text{[overall mean]}}$=0.015, 95% confidence interval, CI = [0.004, 0.025]; *Figure 2, Supplementary file 1*. However, this significant relationship disappeared ($r$=0.0009) once we controlled for species number and effort time (p=0.863, $z$=0.173, $Zr$ = $b_{\text{[overall mean]}}$=0.0009, 95% CI = [–0.0092, 0.0111]); both variables were statistically significant predictors of the effect. As expected, the increase in species number (modelled as the inverse of species number – 3, which is equivalent to sampling error for $Zr$) and effort time on the natural log scale decreased the strength of the relationship (sampling variance: p<0.0001, $z$=140.29; $b_{\text{[sampling variance]}}$=0.0147, 95% CI = [–0.0149, –0.0145]); ln(effort time): p<0.0001, $z$=–183.45, $b_{\text{[ln(effort time)]}}$=0.230, 95% CI = [0.226, 0.233]; marginal $R^2$=5.1% for the model with these two predictors; *Nakagawa and Schielzeth, 2013; Supplementary file 1*). These observations are consistent with the explanation that sampling protocols can create positive artefactual relationships between range and abundance.

Even more remarkably, our meta-analysis suggested that the AOR is likely indistinguishable from zero even with a larger dataset because the observed heterogeneity among effect sizes was very small (i.e. most effect sizes were effectively zero after accounting for sample size). A measure of relative heterogeneity $I^2_{\text{[total]}}$ was 13.5%, meaning that 86.5% of all the observed variation (in *Figure 2*) is due to sampling error, therefore, is neither biological nor ecological (country level; $I^2$=5.5%, $\sigma^2$=0.005; state level: $I^2$=6.3%, $\sigma^2$=0.005; effect-size level; $I^2$=1.5%, $\sigma^2$=0.001); in contrast, the average $I^2_{\text{[total]}}$ across 86 ecological meta-analyses was approximately 92% (*Senior et al., 2016*), making our observed heterogeneity unusually low. Low relative heterogeneity, however, does not necessarily mean absolute heterogeneity is also low (*Borenstein et al., 2017*). We found the absolute heterogeneity, $\sigma^2_{\text{[total]}}$=0.011, approximately one-thirtieth of the heterogeneity ($\sigma^2_{\text{[total]}}$=0.323) found in the previous meta-analysis (*Blackburn et al., 2006*). Also, this is around one-tenth of the average heterogeneity ($\sigma^2_{\text{[total]}}$=0.125; median = 0.105) found in 31 meta-analyses in ecology and evolution (*Yang et al., 2023*). Overall, low relative and absolute heterogeneities indicate that our dataset of 16,562,995 effect sizes does not have much variability left to be explained despite our observations coming from many different locations across the globe and contributed by tens of thousands of individual birdwatchers. Importantly, we emphasise that this combination of zero effect and very small heterogeneity is only expected when a particular phenomenon is not real.

Moreover, our phylogenetic comparative analysis, which accounted for phylogenetic uncertainty (*Nakagawa and De Villemereuil, 2019*), corroborated our meta-analytic results (cf. *Gaston et al., 1997a*). The global range sizes had little predictive power on mean species density (both on log10; p=0.808, $t_{99.6}$ = 0.0227, $b_{\text{[slope]}}$=0.0928, 95% CI = [–0.1615, 0.2068]; *Figure 3, Supplementary file 2*). Taken together, our results provide overwhelming evidence against the fundamental relationship between species range and local abundance, while the results are consistent with this relationship as a sampling artefact. Nevertheless, our results are also consistent with previously published empirical evidence. This is because we have shown that relationships between global species ranges and local counts can be null, strongly negative, or strongly positive, which can be generated primarily by sampling (error) variance (shown in *Figure 2*).

## Discussion

### Lawless macroecology and non-neutral theory: implications

Our results demonstrate clearly that the AOR is not observed in a very large global dataset, with both applied and theoretical ramifications. First, we must reconsider fishing quotas, conservation priorities, and invasive species control strategies based on AORs (cf. *Gaston, 1999a*). Second, it may be futile to pursue all ecological or biological mechanisms proposed for AORs (see *Gaston et al., 1997b; Borregaard and Rahbek, 2010*). We cannot exclude, however, the possibility of AORs occasionally emerging in some restricted areas because there was a small unexplained variance in the meta-analytic dataset. Most notably, our near-zero results with small heterogeneity suggest that contrary to earlier suggestions (*Bell, 2001; Hubbell, 1997*), the spatial neutral model is not a suitable null model of macroecology. Within the neutral theoretical framework, AORs can be broken by local adaptation (an alternative hypothesis) (*Bell, 2001*). If local adaptation were to disrupt a predicted positive

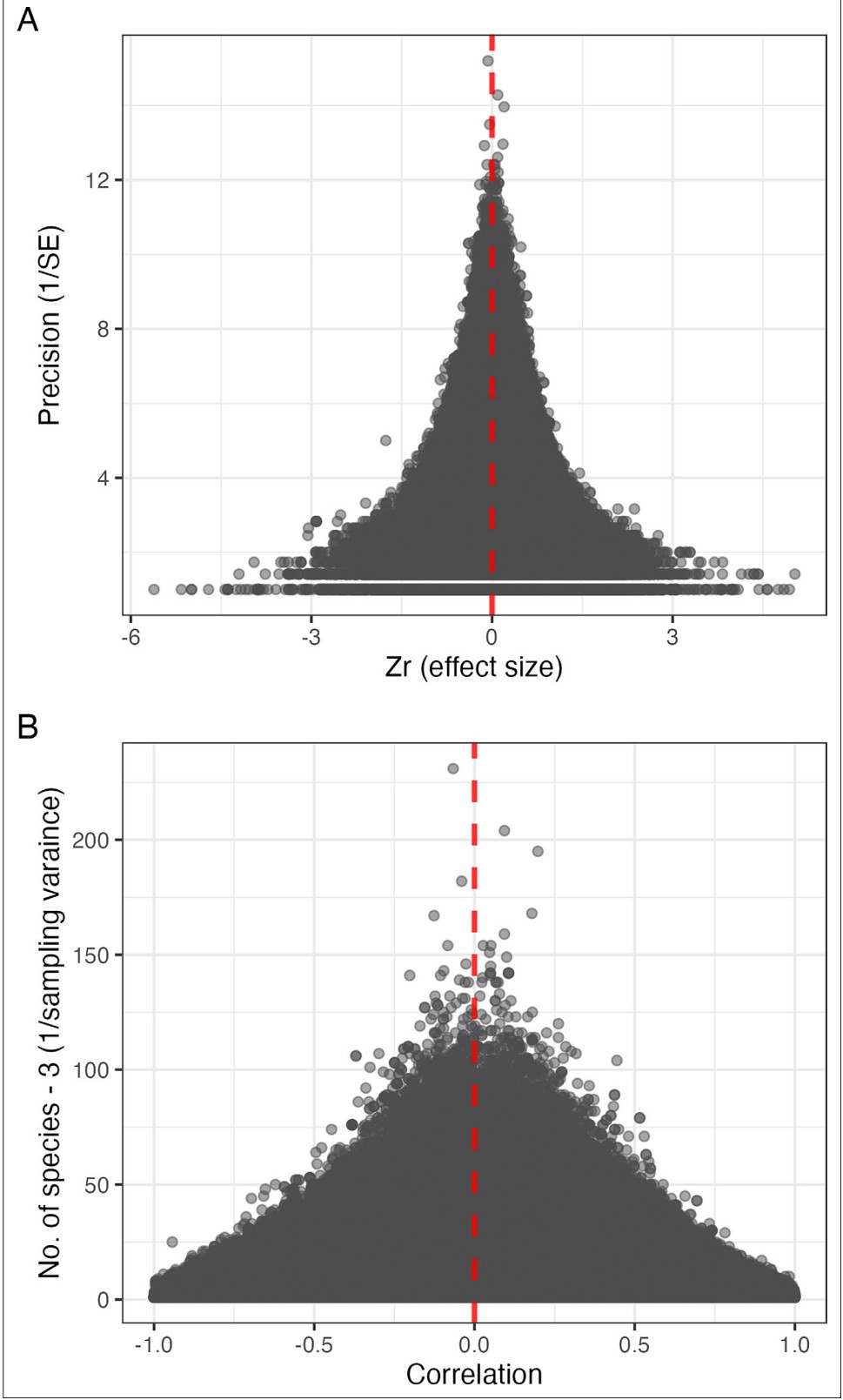

**Figure 2.** Funnel plots. (**A**) The relationship between 16,562,995 effect sizes (Fisher's; *x*-axis) and their precision (the square root of the inverse of the sampling variance; *y*-axis). (**B**) The relationship between 16,562,995 correlations based on 3,005,668,285 observations of 7635 species (Pearson's correlation coefficients; *x*-axis) and the number of species – 3, which is the inverse of the sampling variances for *Zr* (*y*-axis). Both plots consist of data points with the red dashed line indicating zero effect.

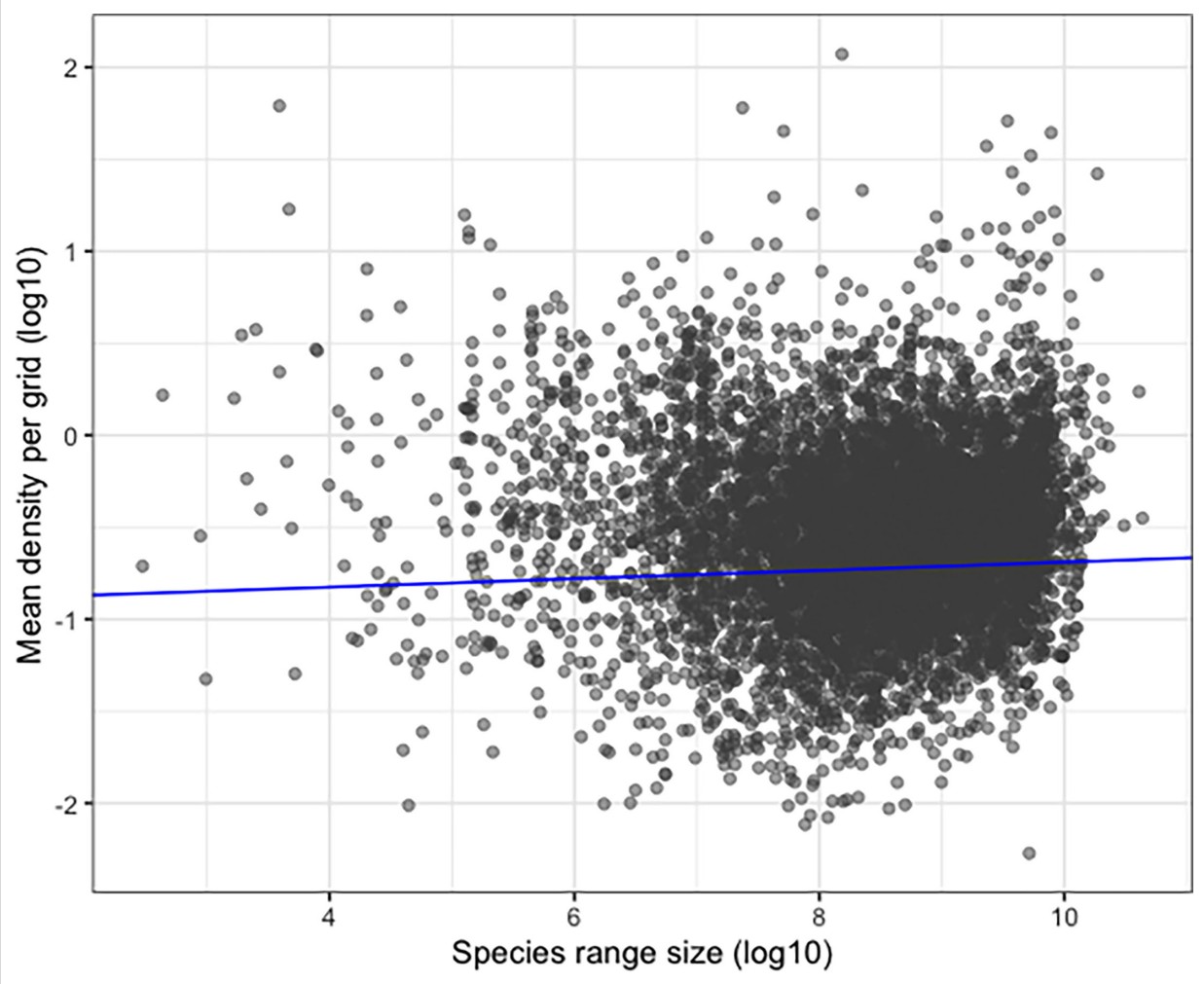

**Figure 3.** The relationship between species average (mean) density and species range size. We calculated the mean density of a species in 5-degree grids where species occurred (*y*-axis), while the species range size (*x*-axis) was estimated by the sum of the percentage occurrence of the species multiplied by the grid size (km²) across all the 575 grids (7464 species). The blue line indicates an average slope line from phylogenetic comparative models with 100 different posterior phylogenetic trees.

relationship (AOR), we would have observed substantial heterogeneity and a reduced relationship, not a near-zero relationship. This is because it is extremely unlikely that local adaptation and neutral processes are in a perfect balance, resulting in an exact-zero relationship with little heterogeneity. In other words, local adaptations are expected to create local specificities and global variability/heterogeneity; our meta-analysis did not find such variability.

We point out that any model of a positive AOR posits a mechanism that connects species range and local abundance on the one hand. Rabinowitz, on the other hand, effectively decoupled these two variables (*Rabinowitz, 1981*), although studies using her framework found positive correlations between species range and local abundance (*Yu and Dobson, 2000*). Our results are consistent with this decoupling. By adding habit specificity to species range and abundance, she suggests seven forms of rarity, which could reflect different underlying macroecological mechanisms; e.g., two forms of rarity are geographically restricted but locally abundant with narrow or broad habitat types. Unlike the world created by the spatial neutral model, our results support many such 'rare' species (*Yu and Dobson, 2000*). In this regard, it is no longer surprising that AORs do not exist. Therefore, we believe Rabinowitz's framework, rather than the AOR, has more empirical support from global-scale patterns of species abundance and provides a useful conceptual structure for future theoretical and applied work, although this line of work is still limited.

## Potential limitations of current work and future work

The primary aim of our study was to demonstrate the potential of large-scale citizen science datasets, such as eBird, to revisit and refine longstanding macroecological relationships. These data, with their global coverage and unprecedented spatial resolution, offer unique opportunities to explore broad-scale patterns beyond the scope of traditional, localised studies (e.g. *Ten Caten et al., 2022*). Although our findings challenge some long-held assumptions about the consistency of the AOR, our work only deals with interspecific AORs among birds, synthesising observations from potentially heterogeneous locations, ecological contexts, and data quality. Therefore, we hope this work serves as we view this study as a foundation for further investigations that utilise such comprehensive datasets.

Future studies could delve deeper into specific ecological factors that may shape interspecific AORs if they do exist. For instance, investigating how islands might influence abundance-density patterns could shed light on density release, where species on islands achieve higher densities due to reduced competition and predation. Additionally, exploring the impact of latitude and climate, such as how Rapoport's rule may lead to more extensive ranges and population sizes in temperate regions (*Gaston et al., 2000*), could provide valuable insights into the variability of AORs across geographic and climatic gradients. Similarly, examining species-specific traits, including body size or wing morphology, may uncover correlations with range size and abundance. We further acknowledge that we did not account for anthropogenic changes in populations or range sizes in our analyses and, therefore, we included alien species without separating them from native ones. More precise range-size estimates would also improve the accuracy of AOR assessments since species range data are often overestimated due to the failure to capture gaps in actual distributions (*Ocampo-Peñuela et al., 2016*).

Beyond these biological and ecological factors, methodological refinements using citizen science data are also needed (*Callaghan et al., 2019*). Our approach, which relies on relative abundance measures, provides a starting point. While our approach relies on relative abundance measures as a starting point, more sophisticated methods are needed to account for known biases (e.g. differences in species detectability, observer experience) in citizen science data so as to enhance the precision of future macroecological studies. We therefore encourage further work to explore novel analytical approaches and statistical frameworks designed to handle these inherent biases, including variation in both observer effort and detectability across species and habitats. Such improvements should help clarify the conditions under which AORs may emerge, remain weak, or are fully decoupled.

## A credibility revolution in ecology beyond biases and crises

Whilst we provided an explanation for the non-existence of AOR with our work's limitations in mind, it still feels hard to comprehend the extent of overestimation in the previous meta-analysis ($r$=0.58 for all taxa; $r$=0.74 for birds) (*Blackburn et al., 2006*; *Blackburn and Gaston, 2009*). We have shown that some bias may be due to sampling bias. However, we speculate that much of the overestimation originates from publication bias and confirmation bias, which is supported by mounting evidence from meta-research studies (*Holman et al., 2015*; *van Wilgenburg and Elgar, 2013*; *Yang et al., 2023*). Although we do not have direct evidence, our eBird datasets are free from these two types of biases (i.e. birdwatchers generally do not think of the macroecological patterns that would later be tested with the data they submit), while the literature-based meta-analyses are not. Regarding publication bias, the original meta-analysis of AOR states, 'the fail-safe number indicates that more than half a million unpublished null results would be required to nullify an effect of this magnitude' (*Blackburn et al., 2006*). Indeed, we provided much more than half a million null effects to reach our null conclusion (*Figure 2*). However, we should note that large datasets like eBird have other biases than publication or confirmation biases (*Callaghan et al., 2017*). For example, it is possible that by excluding checklists with a single 'X' (see Materials and methods), we are preferentially removing abundant species as birdwatchers may report 'X' for more common species with high abundances.

A recent study re-examining 86 ecological and evolutionary meta-analyses demonstrated a 23% reduction in overall effects due to publication bias, turning 33 of 50 statistically significant meta-analytic conclusions (66%) into non-significant (*Yang et al., 2023*). Similarly, a study examining 83 topics in life sciences showed that the effect size of non-blind studies, which are at risk of confirmation bias, was twice as large as blinded counterparts protected against confirmation bias (*Holman et al., 2015*). Meta-research on behavioural ecology identified 79 studies on nestmate recognition,

23 of which were conducted blind (*van Wilgenburg and Elgar, 2013*). Non-blind studies confirmed a hypothesis of no aggression towards nestmates nearly three times more often. It is possible that confirmation bias was at play in earlier AOR studies.

We finish with an intriguing parallel topic to AORs in psychology, where the current replication crisis started (*Aarts, 2015*; *Eronen and Bringmann, 2021*). There have been over 100 studies, and many theoretical models support the hypothesis of 'ego depletion', where self-control is a finite resource, so self-control will decrease once it is exerted (*Inzlicht and Schmeichel, 2012*). The first meta-analysis of ego depletion, like AOR, suggested very strong support for it (standardised mean difference, or $d$=0.62). Yet, a subsequent multi-lab replication found that ego depletion does not exist and is so weak as to be negligible ($d$=0.04) (*Hagger et al., 2016*). Indeed, a series of multi-lab replications has indicated that several psychological phenomena, which were once believed to be real beyond a reasonable doubt, are too weak to be useful or are non-existent (*Kvarven et al., 2020*). In ecology, recently collated large datasets collected for non-hypothesis-driven purposes offer a unique opportunity to revisit and retest longstanding ideas.

Taken together, we call for re-examining all ecological laws, rules, and patterns, as very few topics are free from sampling, confirmation, and publication biases (cf. *Hughes et al., 2021*). To counter such biases, we urgently require a 'credibility revolution', a more optimistic name for a replication crisis, turning this crisis into an opportunity to improve science. A credibility revolution in ecology, like in psychology, needs to embrace non-traditional methods to avoid confirmation and publication bias, such as pre-registration (*Nosek et al., 2018*), registered reports (*Chambers and Tzavella, 2022*), prospective and living meta-analyses, open synthesis communities (*Nakagawa et al., 2020*), and big-team-science collaborations (*Coles et al., 2022*) involving community (citizen) scientists (*Callaghan et al., 2019*).

## Materials and methods
### Quantifying AOR at the local scale

We used the eBird dataset (*Sullivan et al., 2014*; *Sullivan et al., 2009*) to assess the relationship between local-scale abundance and occupancy (i.e. global range size). eBird, launched in 2002 by the Cornell Lab of Ornithology, is a global citizen science project that enlists volunteer birdwatchers to submit 'checklists' of birds seen and/or heard while birdwatching. Data undergo a semi-automated filtering process before being entered into the dataset, and expert reviewers additionally review species (or counts of species) that surpass preset filters before being accepted into the dataset (*Gilfedder et al., 2019*). Importantly, birdwatchers must indicate whether they are submitting a 'complete' checklist representing all birds that an individual birdwatcher was able to identify during their birdwatching outing. Further, birdwatchers can either submit the count of a species during their birding, or they can submit an 'X' to signify that a species was present but not estimate the number of individuals present during their birdwatching outing.

We downloaded the eBird basic dataset (version ebd_rel-May2020) and considered all eBird checklists between 1 January 2005 and 31 May 2020. We then performed some quality assurance, applying an additional set of filters to the data, potentially removing any 'outliers' that could produce undue leverage on our results. The following filtering was completed (sensu *Callaghan et al., 2017*; *Johnston et al., 2021*). We only included: (1) complete checklists; (2) checklists that were <240 min and >5 min; (3) checklists that travelled <5 km; and (4) checklists that travelled <500 ha. Because birdwatchers will sometimes use an 'X' to signify presence, and this is most likely to happen for more abundant species, we excluded any checklist that had at least an 'X' on it, as this could potentially influence the correlation between the abundance of a species and range size by disproportionately removing the most abundant species from the correlation. This exclusion aimed to ensure that correlations between local abundance and range size were not distorted by the lack of abundance data for highly observable, widespread species, and providing all species on a checklist with an abundance estimate maximises the interpretability of the relative abundance measure in our work. We further only considered checklists that had at least 10 species recorded on them, and a correlation test was performed only if we had range size data (see below) for a minimum of four species on the checklist.

We used range size maps from *BirdLife International, 2023*, using their global range, ignoring the differences between resident and breeding ranges. We chose to use the global range because of the

difficulty of defining species occupancies using grid cells (i.e. almost infinite ways of defining occupancy) and the importance of using the entire species distribution range pointed out by earlier studies (e.g. *Bock and Ricklefs, 1983*) due to sampling artefacts. When an eBird checklist met the aforementioned criteria, we performed a correlation test using Pearson's correlation coefficient from the *cor. test* function in R (*R Development Core Team, 2023*). Both the counts of every species and the range size were log-transformed before estimating a correlation (for a workflow, see *Figure 1*). We obtained 16,562,995 correlations based on 3,005,668,285 individual bird observations, including 7635 species. We note that we conducted all computational and statistical work using R, and we created plots using the R package *ggplot2* (*Wickham, 2016*), patchwork (*Pedersen, 2022*), and their dependencies.

## Meta-analysis of Big Data

We transformed correlations between species abundance and range into Fisher's $Z$ or $Zr$ to unbound and calculated sampling variance for each $Zr$ value; note that the inverse of the sampling variance of $Zr$ is $N$ (the number of species in a checklist) – 3 (see *Figure 1*). We used the R package, *asreml* (*Nakagawa and Santos, 2012*), to run a multilevel random effects model (*Nakagawa and Santos, 2012*); note that *asreml* is a commercial package, so it is not free. Our large meta-analyses with ~17 million effect sizes were only able to run with *asreml* given the computational time required for such a large dataset. We had 'country' (245 levels) and state code (2871 levels) as random factors in the model to control for non-independence. In addition, to quantify the variance component for these two clustering factors and also at the level of effect sizes (16,562,995 levels), we modelled 'units' (the effect size level random effect or residuals) in the *asreml* function with 'the number of species – 3' as the 'weights' argument and *asr_gaussian(dispersion = 1)* as the 'family' argument. We also obtained the multilevel versions of $I^2$ (*Senior et al., 2016*; *Higgins and Thompson, 2002*) to obtain relative heterogeneity for our meta-analytic model (*Supplementary file 1*; also, all models used in this study are summarised in *Supplementary file 3*).

To gauge the impacts of potential biases, we fitted two moderators: (1) the $z$-transformed version of ln(checklist duration) as a surrogate for the amount of effort for observation and (2) sampling variance, which is usually used to detect publication bias, more specifically, small study bias where effect sizes from small studies can create 'funnel asymmetry', creating bias in meta-analytic overall mean (*Nakagawa et al., 2022*; *Figure 2*). We ran two uni-moderator models and one multi-moderator model with both moderators (three meta-regression models in total; *Supplementary file 1*). We estimated the multilevel model versions of $R^2$ (*Nakagawa and Schielzeth, 2013*).

## Quantifying the AOR at the macro-scale

To corroborate our local-level analysis described above, we quantified an additional macro-scale analysis of the relationship between abundance (i.e. density) and occupancy (global range size). For this, we used data from a recently published analysis of global abundances for birds within 5-degree grid cells (*Callaghan et al., 2021*). This dataset was derived by integrating expert-derived abundance measures with a large, less structured global citizen science dataset using a multiple-imputation technique to estimate density within 5-degree grids for 9700 bird species (575 grids). We used these predicted density estimates from each grid cell and, for each species, took the mean of all density estimates in the grid cells for which a species was found. This mean density was then our measure of macro-scale abundance. For our measure of occupancy, we used a summation of all range sizes for the grids a species was found in, calculated by using range maps from BirdLife International focusing on the entire extent of a species' extant range, ignoring the effect of transient species. Our analysis incorporated a total of 7464 species of bird species, corresponding to the species for which we had both range maps and estimated density, along with phylogenetic information included in *Jetz et al., 2012*.

## Phylogenetic comparative analysis

To statistically test whether there was an effect of abundance and occupancy at the macro-scale, we used phylogenetic comparative analysis. This analysis also addresses the issue of positive interspecific AORs potentially arising from not accounting for phylogenetic relatedness among species examined (*Gaston et al., 2000*). We used avian phylogeny from *Jetz et al., 2012*, and analysed 100 phylogenetic trees using the R function *phylolm* (*Ho and Ané, 2014*). Resulting estimates from the 100 models were merged using Rubin's rules, as described in *Nakagawa and De Villemereuil, 2019*, to obtain

current estimates and errors that accounted for phylogenetic uncertainty (*Supplementary file 2*); we implemented this procedure using the R function *miInference* from the *norm2* package (*Schafer, 2021*).

## Acknowledgements

We thank Szymek Drobniak for helping run large meta-analytic models on a cluster computer system. We are also grateful to Malgorzata Lagisz, Tim Parker, Daniel Noble, Tim Blackburn, Diana Bowler, and Luís Borda de Agua for their comments on earlier versions of this manuscript.

## Additional information

### Funding

| Funder | Grant reference number | Author |
|---|---|---|
| Australian Research Council | DP210100812 | Shinichi Nakagawa |

The funders had no role in study design, data collection and interpretation, or the decision to submit the work for publication.

### Author contributions

Shinichi Nakagawa, Conceptualization, Investigation, Visualization, Methodology, Writing – original draft, Project administration, Writing – review and editing; William K Cornwell, Conceptualization, Investigation, Methodology, Writing – original draft, Writing – review and editing; Corey T Callaghan, Conceptualization, Investigation, Visualization, Methodology, Writing – original draft, Writing – review and editing

### Author ORCIDs

Shinichi Nakagawa ⬤ https://orcid.org/0000-0002-7765-5182
William K Cornwell ⬤ https://orcid.org/0000-0003-4080-4073
Corey T Callaghan ⬤ http://orcid.org/0000-0003-0415-2709

Reviewer #1 (Public review): https://doi.org/10.7554/eLife.95857.3.sa1
Author response https://doi.org/10.7554/eLife.95857.3.sa2

## Additional files

### Supplementary files

MDAR checklist

Supplementary file 1. Statistical results from meta-analytic models.

Supplementary file 2. Statistical results from comparative analysis.

Supplementary file 3. Mathmatical formulations for meta-analytic models.

### Data availability

All data, code, and materials are available online unless they are too large to be archived (https://github.com/itchyshin/AORs copy archived at *Nakagawa, 2025*) and they are archived in a public repository, Zenodo (https://doi.org/10.5281/zenodo.14019900).

The following dataset was generated:

| Author(s) | Year | Dataset title | Dataset URL | Database and Identifier |
|---|---|---|---|---|
| Nakagawa S | 2024 | itchyshin/AORs: Abundance_Ocupancy_Relationship | https://doi.org/10.5281/zenodo.14019900 | Zenodo, 10.5281/zenodo.14019900 |

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
