## [Editor Report · eLife assessment]

This study offers a **useful** discussion of the well-accepted abundance-occupancy relationship in macroecology. While using the ebird large dataset to revisit the theme is interesting, multiple unresolved confounding factors exist, leaving the results **inadequate** to overturn the repeatedly confirmed abundancy-occupancy relationship.

---

## [Referee Report · Reviewer #1 (Public review)]

Summary:

This article presents an analysis that challenges established abundance-occupancy relationships (AORs) by utilizing the largest known bird observation database. The analysis yields contentious outcomes, raising the question of whether these findings could potentially refute AORs.

Strengths:

The study employed an extensive aggregation of datasets to date to scrutinize the abundance-occupancy relationships (AORs).

Weaknesses:

The authors should thoroughly address the correlation between checklist data and global range data, ensuring that the foundational assumptions and potential confounding factors are explicitly examined and articulated within the study's context.

In the revision, the authors have refined their findings to birds and provided additional clarifications and discussion. However, the primary concerns raised by reviewers remain inadequately addressed. My main concern continues to be whether testing AOR at a global scale is meaningful given the numerous confounding factors involved. With the current data and analytical approach, these confounders appear inseparable. The study would be significantly strengthened if the authors identified specific conditions under which AORs are valid.

---

## [Author Response]

The following is the authors’ response to the original reviews

**Public Reviews:**

**Reviewer #1 (Public Review):**
Summary:This article presents a meta-analysis that challenges established abundance-occupancy relationships (AORs) by utilizing the largest known bird observation database. The analysis yields contentious outcomes, raising the question of whether these findings could potentially refute AORs.

We thank the Reviewer for their positive comments.

Strengths:The study employed an extensive aggregation of datasets to date to scrutinize the abundance-occupancy relationships (AORs).

We thank the Reviewer for their positive comments.

Weaknesses:While the dataset employed in this research holds promise, a rigorous justification of the core assumptions underpinning the analytical framework is inadequate. The authors should thoroughly address the correlation between checklist data and global range data, ensuring that the foundational assumptions and potential confounding factors are explicitly examined and articulated within the study's context.

We thank the Reviewer for these comments. We agree that more justification and transparency is needed of the core assumptions that form the foundation of our methods. In our revised version, we have taken the following steps to achieve this:

- Altered the title to be more explicit about the core assumptions, which now reads: “Local-scale relative abundance is decoupled from global range size”

- We have added more details on why and how we treat global range size as a measure of ‘occupancy.’

- We have added a section that discusses the limitations of using eBird relative abundance

**Reviewer #2 (Public Review):**
Summary:The goal is to ask if common species when studied across their range tend to have larger ranges in total. To do this the authors examined a very large citizen science database which gives estimates of numbers, and correlated that with the total range size, available from Birdlife. The average correlation is positive but close to zero, and the distribution around zero is also narrow, leading to the conclusion that, even if applicable in some cases, there is no evidence for consistent trends in one or other direction.

We thank the Reviewer for these comments.

Strengths:The study raises a dormant question, with a large dataset.

We thank the Reviewer for these comments. We intended to take a longstanding question and attempt to apply novel datasets that were not available mere decades ago. While we do not imply that we have ‘solved’ the question, we hope this work highlights the potential for further interrogation using these large datasets.

Weaknesses:This study combines information from across the whole world, with many different habitats, taxa, and observations, which surely leads to a quite heterogeneous collection.

We agree that there is a heterogeneous collection of data across many habitats, taxa, and observations. However, rather than as a weakness, we see this as a significant strength. Our work assumes we are averaging over this variability to assess for a large-scale pattern in the relationship - something that was potentially a limitation of previous work, as these large datasets were often focused on particular contexts (e.g., much work focused solely on the UK), which we believe could limit some of the generalizability of the previous work. However, the reviewer makes a fair point in regard to the heterogeneity of data collection. We have now added some text in the discussion which is explicit about this - see the new section named “Potential limitations of current work and future work –-although our findings challenge some long-held assumptions about the consistency of the abundance-occupancy relationship, our work only deals with interspecific AORs among birds, synthesizing observations of potentially heterogeneous locations, context and quality”.

First, scale. Many of the earlier analyses were within smaller areas, and for example, ranges are not obviously bounded by a physical barrier. I assume this study is only looking at breeding ranges; that should be stated, as 40% of all bird species migrate, and winter limitation of populations is important. Also are abundances only breeding abundances or are they measured through the year? Are alien distributions removed?Second, consider various reasons why abundance and range size may be correlated (sometimes positively and sometimes negatively) at large scales. Combining studies across such a large diversity of ecological situations seems to create many possibilities to miss interesting patterns. For example:(1) Islands are small and often show density release.

See comment below.

(2) North temperate regions have large ranges (Rapoport's rule) and higher population sizes than the tropics.

See comment below.

(3) Body size correlates with global range size (I am unsure if this has recently been tested but is present in older papers) and with density. For example, cosmopolitan species (barn owl, osprey, peregrine) are relatively large and relatively rare.

See comment below.

(4) In the consideration of alien species, it certainly looks to me as if the law is followed, with pigeon, starling, and sparrow both common and widely distributed. I guess one needs to make some sort of statement about anthropogenic influences, given the dramatic changes in both populations and environments over the past 50 years.

See comment below. We also added a sentence in the methods that highlighted we did not remove alien ranges and provided reasons why. Still, we do acknowledge the dramatic changes in populations and environments over the past 50 years (see the new section “Potential limitations of current work and futur work”)

(5) Wing shape correlates with ecological niche and range size (e.g. White, American Naturalist). Aerial foraging species with pointed wings are likely to be easily detected, and several have large ranges reflecting dispersal (e.g. barn swallow).

We agree that all of the points above are interesting data explorations. As said above, our main purpose was to highlight the potential for further interrogation using these large datasets. However, we have added some additional text in the discussion that explicitly mentions/encourages these additional data explorations. We hope people will pick up on the potential for these data and explore them further.

Third, biases. I am not conversant with ebird methodology, but the number appearing on checklists seems a very poor estimate of local abundance. As noted in the paper, common species may be underestimated in their abundance. Flocking species must generate large numbers, skulking species few. The survey is often likely to be in areas favorable to some species and not others. The alternative approach in the paper comes from an earlier study, based on ebird but then creating densities within grids and surely comes with similar issues.

We agree that if we were interested in the absolute abundance of a given species, the local number on an eBird checklist would be a poor representation. However, our study aims not to estimate absolute abundance but to examine relative abundance among species on each checklist. By focusing on relative abundance, we leverage eBird data's strengths in detecting the presence and frequency of species across diverse locations and times, thereby capturing community composition trends that can provide meaningful insights despite individual checklist biases. This approach allows us to assess the comparative prominence of species in the community as reported by the observer, providing a consistent metric of relative abundance. Despite detectability biases, the structure of eBird checklists reflects the observer’s encounter rates with each species under similar conditions, offering a valuable snapshot of relative species composition across sites and times. The key to our assumption is that these biases discussed are not directional and, therefore, random throughout the sampling process, which would translate to no ‘real’ bias in our effect size of interest.

Range biases are also present. Notably, tropical mountain-occupying species have range sizes overestimated because holes in the range are not generally accounted for (Ocampo-Peñuela et al., Nature Communications). These species are often quite rare, too.

We thanks the reviewer for pointing to this issue and reference. We included a discussion on these biases in our limitations section and reference Ocampo-Peñuela et al. to emphasize the need for improved spatial resolution in range data for more accurate AOR assessments.”More precise range-size estimates would also improve the accuracy of AOR assessments, since species range data are often overestimated due to the failure to capture gaps in actual distributions ”

Fourth, random error. Random error in ebird assessments is likely to be large, with differences among observers, seasons, days, and weather (e.g. Callaghan et al. 2021, PNAS). Range sizes also come with many errors, which is why occupancy is usually seen as the more appropriate measure.If we consider both range and abundance measurements to be subject to random error in any one species list, then the removal of all these errors will surely increase the correlation for that list (the covariance shouldn't change but the variances will decrease). I think (but am not sure) that this will affect the mean correlation because more of the positive correlations appear 'real' given the overall mean is positive. It will definitely affect the variance of the correlations; the low variance is one of the main points in the paper. A high variance would point to the operation of multiple mechanisms, some perhaps producing negative correlations (Blackburn et al. 2006).

We agree random errors can affect estimates, but as we wrote above, random errors, regardless of magnitudes, would not bias estimates. After accounting for sampling error (a part of random errors), little variance is left to be explained as we have shown in the MS. This suggests that many of the random errors were part of the sampling errors. And this is where meta-analysis really shines.

On P.80 it is stated: "Specifically, we can quantify how AOR will change in relation to increases in species richness and sampling duration, both of which are predicted to reduce the magnitude of AORs" I haven't checked the references that make this statement, but intuitively the opposite is expected? More species and longer durations should both increase the accuracy of the estimate, so removing them introduces more error? Perhaps dividing by an uncertain estimate introduces more error anyway. At any rate, the authors should explain the quoted statement in this paper.It would be of considerable interest to look at the extreme negative and extreme positive correlations: do they make any biological sense?

Extremely high correlations would not make any biological sense if these observations were based on large sample sizes. However, as shown in Figure 2, all extreme correlations come from small sample sizes (i.e., low precision), as sampling theory expects (actually our Fig 2 a text-book example of the funnel shape). Therefore, we do not need to invoke any biological explanations here.

Discussion:I can see how publication bias can affect meta-analyses (addressed in the Gaston et al. 2006 paper) but less easily see how confirmation bias can. It seems to me that some of the points made above must explain the difference between this study and Blackburn et al. 2006's strong result.

We agree. Now, we extended an explanation of why confirmation bias could result in positive AOR. Yet, we point out confirmation bias is a very common phenomena which we cite relevant citations in the original MS. The only way to avoid confirmation bias is to conduct a study blind but this is not often possible in ecological work.

“Meta-research on behavioural ecology identified 79 studies on nestmate recognition, 23 of which were conducted blind. Non-blind studies confirmed a hypothesis of no aggression towards nestmates nearly three times more often. It is possible that confirmation bias was at play in earlier AOR studies.”

Certainly, AOR really does seem to be present in at least some cases (e.g. British breeding birds) and a discussion of individual cases would be valuable. Previous studies have also noted that there are at least some negative and some non-significant associations, and understanding the underlying causes is of great interest (e.g. Kotiaho et al. Biology Letters).

We agree. And yes, we pointed out these in our introduction.

**Reviewer #3 (Public Review):**
Summary:This paper claims to overturn the longstanding abundance occupancy relationship.Strengths:(1) The above would be important if true.(2) The dataset is large.

We have clarified this point by changing the title to emphasize that we do not suggest overturning AORs entirely but instead provide a refined view of the relationship at a global scale. Our results suggest a weaker and more context-dependent AOR than previously documented. We hope our revised title and additional clarifications in the text convey our intent to contribute to a more nuanced understanding rather than a whole overturning of the AOR framework.

Weaknesses:(1) The authors are not really measuring the abundance-occupancy relationship (AOR). They are measuring abundance-range size. The AOR typically measures patches in a metapopulation, i.e. at a local scale. Range size is not an interchangeable notion with local occupancy.

We have refined this in our revision to be more explicitly focused on global range size. However, we note that the classic paper by Bock and Richlefs (1983, Am Nat) also refers to global (species entire) range size in the context of the AOR. Importantly, Bock and Richlefs pointed out the importance of using species’ entire ranges; without such uses, there will be sampling artifacts creating positive AORs when using arbitrary geographical ranges, which were used in some studies of AORs. So we highlight that our work is well in line with the previous work, allowing us to question the longstanding macroecological work. One of the issues of AOR has been how to define occupancy and global range size, which provides a relatively ambiguous measure, which is why we used this measure.

(2) Ebird is a poor dataset for this. The sampling unit is non-standard. So abundance can at best be estimated by controlling for sampling effort. Comparisons across space are also likely to be highly heterogenous. They also threw out checklists in which abundances were too high to be estimated (reported as "X"). As evidence of the biases in using eBird for this pattern, the North American Breeding Bird Survey, a very similar taxonomic and geographic scope but with a consistent sampling protocol across space does show clear support for the AOR.

Yes, we agree the sampling unit is non-standard. However, this is a significant strength in that it samples across much heterogeneity (as discussed in response to Reviewer 2, above). We were interested in relative abundance and not direct absolute abundance per se, which is accurate, especially since we did control for sampling effort.

We appreciate the reviewer’s attention to our data selection criteria. We excluded checklists containing ‘X’ entries to minimize biases in our abundance estimates. The 'X' notation is often used for the most common species, reflecting the observer's identification of presence without specifying a count. This approach was chosen to avoid disproportionately inflating presence data for these abundant species, which could distort the relative abundance calculations in our analysis. By excluding such checklists, we aimed to retain consistency and ensure that local abundance estimates were representative across all species on each checklist. We have revised our manuscript to clarify this methodological choice and hope this explanation addresses the reviewer’s concern. We modified our text in the methods to make the entries ‘X’ clearer (see the Method section).

(3) In general, I wonder if a pattern demonstrated in thousands of data sets can be overturned by findings in one data set. It may be a big dataset but any biases in the dataset are repeated across all of those observations.

Overturning a major conclusion requires careful work. This paper did not rise to this level.

We appreciate the reviewer’s caution regarding broad conclusions based on a single dataset, even one as large as eBird. Our intention was not to definitively overturn the abundance-occupancy relationship (AOR) but to re-evaluate it with the most extensive and globally representative dataset currently available. We recognise that potential biases in citizen science data, such as observer variation, may influence our findings, and we have taken steps to address these in our methodology and limitations sections. We see this work as a contribution to an ongoing discourse, suggesting that AOR may be less universally consistent than previously believed, mainly when tested with large-scale citizen science data. We hope this study will encourage additional research that tests AORs using other expansive datasets and approaches, further refining our understanding of this classic macroecological relationship. However, we have left our broad message about instigating credible revolution and also re-examining ecological laws.

**Recommendations for the authors:**

**Reviewer #1 (Recommendations For The Authors):**
(1) The investigation focuses solely on interspecific relationships among birds; thus, the extrapolation of these conclusions to broader ecological contexts requires further validation.

We have now added this point to our new section: “Although our findings challenge some long-held assumptions about the consistency of the abundance-occupancy relationship, our work only deals with interspecific AORs among birds, so we hope this work serves as a foundation for further investigations that utilize such comprehensive datasets.”

(2) The rationale for combining data from eBird - a platform predominantly representing individual observations from urban North America - with the more globally comprehensive BirdLife International database needs to be substantiated. The potential underrepresentation of global abundance in the eBird checklist data could introduce a sampling bias, undermining the foundational premises of AORs.

We agree with the limitation of ebird sampling coverage, but it should not bias our results. In statistical definitions, bias is directional, and if not directional, it will become statistical noise, making it difficult to detect the signal. In fact, our meta-analyses adjust what statisticians call sampling bias and it is the strength of meta-analysis.

(3) In the full mixed-effect model, checklist duration and sampling variance (inversely proportional to sample size N) are treated as fixed effects. However, these variables are likely to be negatively correlated, which could introduce multicollinearity, inflating standard errors and diminishing the statistical significance of other factors, such as the intercept. This calls into question the interpretation of insignificance in the results.

Multicollinearity is an issue with sample sizes. For example, with small datasets, correlations of 0.5 could be an issue, and such an issue would usually show up as a large SE. We do not have such an issue with ~ 17 million data points. Please refer to this paper.

Freckleton, Robert P. "Dealing with collinearity in behavioural and ecological data: model averaging and the problems of measurement error." *Behavioral Ecology and Sociobiology* 65 (2011): 91-101.

(4) The observed low heterogeneity may stem from discrepancies in sampling for abundance versus occupancy, compounded by uncertainties in reporting behavior.

If we assume everybody underreports common species or overreports rare species, this could happen. However, such an assumption is unlikely. If some people report accurately (but not others), we should see high heterogeneity, which we do not observe. We have touched upon this point in our original MS.

(5) The contribution and implementation of phylogenetic comparative analysis remain ambiguous and were not sufficiently clarified within the study.

We need to add more explanation for the global abundance analysis

“To statistically test whether there was an effect of abundance and occupancy at the macro-scale, we used phylogenetic comparative analysis. This analysis also addresses the issue of positive interspecific AORs potentially arising from not accounting for phylogenetic relatedness among species examined ”

(6) The use of large N checklists could skew the perceived rarity or commonality of species, potentially diminishing the positive correlation observed in AORs. A consistent observer effect could lead to a near-zero effect with high precision.

Regardless of the number of N species in checklists (seen in Fig 2), correlations are distributed around zero. This means there is nothing special about large N checklists.

(7) The study should acknowledge and discuss any discrepancies or deviations from previous literature or expected outcomes.

We felt we had already done this as we discussed the previous meta-analysis and what we expected from this meta-analysis. Nevertheless, we have added some relevant sentences in the new version of MS.

In addition to these major points, there are several minor concerns:(1) Figure 2B lacks discussion, and the metric for the number of observations is not clarified. Furthermore, the labeling of the y-axis appears to be incorrect.

Thank you very much for pointing out this shortcoming. Now, the y-axis label has been fixed and we mention 2B in the main text.

(2) The study should provide a clear, mathematical expression of the multilevel random effect models for greater transparency.

Many thanks for this point, and now we have added relevant mathematical expressions in Table S6.

(3) On Line 260, the term "number of species" should be refined to "number of species in a checklist," ideally represented by a formula for precision.

This ambiguity has been mended as suggested.

Please provide the data and R code linked to the outputs.

The referee must have missed the link (https://github.com/itchyshin/AORs) in our original MS. In addition to our GitHub repository link, we now have added a link to our Zenodo repository (https://doi.org/10.5281/zenodo.14019900).

**Reviewer #3 (Recommendations For The Authors):**
The authors cite Rabinowitz's 7 forms of rarity paper as a suggestion that previous findings also break the AOR. In fact empirical studies of the 7 forms of rarity typically find that all three forms of rareness vs commonness are heavily correlated (e.g. Yu & Dobson 2000).

We thank the reviewer for drawing attention to Yu & Dobson (2000) and similar studies that find positive correlations among the axes of rarity. Ref 3 is correct in that Rabinowitz’s (1981) framework does not require that local abundance and geographic range size be uncorrelated for every species; instead, it highlights conceptual scenarios where a species may be common locally yet have a restricted distribution (or vice versa).

Empirical analyses such as Yu & Dobson (2000) show that, on average, these axes can be correlated, which *may* align with conventional AOR findings in some taxonomic groups. However, Rabinowitz’s key insight was that exceptions do occur, so these exceptions demonstrate that strong positive AORs may not be universally applicable. Our results do not claim that Rabinowitz’s framework “breaks” the AOR outright; instead, we use it to underscore that local abundance can, in principle, be “decoupled” from global occupancy. Whether the correlation found by Yu & Dobson (2000) implies a positive AOR, requires a detailed simulation study, which is an interesting avenue for future research.

Thus, citing Rabinowitz serves to highlight the potential heterogeneity and complexity of abundance–occupancy relationships rather than to refute every positive correlation reported in the literature. Our findings suggest that when examined at large spatiotemporal scales (with unbiased sampling), the overall AOR signal may be less robust than traditionally believed. This is consistent with Rabinowitz’s view that local abundance and global range can vary along independent axes. Now we added

“Although studies using her framework found positive correlations between species range and local abundance.”